# Expression of E-Cadherin and N-Cadherin in the Endocervix as a Predictive Factor in Patients with Endometrial Cancer

**DOI:** 10.3390/ijms25063547

**Published:** 2024-03-21

**Authors:** Karolina Frąszczak, Bartłomiej Barczyński, Bożydar Tylus, Wiesława Bednarek

**Affiliations:** 11st Chair and Department of Oncological Gynaecology and Gynaecology, Medical University in Lublin, 20-081 Lublin, Poland; karolina.fraszczak@umlub.pl (K.F.); wbed@wp.pl (W.B.); 2Department of Obstetrics and Pathology of Pregnancy, Medical University in Lublin, 20-081 Lublin, Poland; 3Gynaecological and Obstetrics Department, Provincial Specialist Hospital, 21-500 Biała Podlaska, Poland; tylusb@gmail.com

**Keywords:** E-cadherin, endometrial cancer, epithelial–mesenchymal transition, N-cadherin

## Abstract

Endometrial cancer (EC) is the most common gynecological malignancy. This study aimed to evaluate the expression of E-cadherin and N-cadherin in primary endometrial lesions and the endocervix in patients with EC to identify noninvasive predictive factors. In this single-center retrospective study, data on 101 patients who underwent surgery for EC were collected. The immunohistochemical expression of E-cadherin and N-cadherin was assessed depending on the tumor grade, location, and cell differentiation. Correlations between E-cadherin and N-cadherin levels in the endocervix and the primary tumor were determined. The degree of histological tumor differentiation significantly affected E-cadherin expression (*p* = 0.04) but had no impact on N-cadherin levels. In type II EC, the expression of both cadherins in the tumor tissue differed from their endocervical levels. The expression of E-cadherin differed significantly between the endocervix (*p* < 0.001) and the tumor (*p* = 0.001), depending on the type of EC. The expression of E-cadherin was related to the N-cadherin level only in the endocervix in patients with type II EC (*p* = 0.02). E-cadherin and N-cadherin were expressed in the endocervix in patients with EC. The expression of cadherins, determined during cervical cytology, may be a valuable clinical marker of EC.

## 1. Introduction

Endometrial cancer (EC) is the most common gynecological malignancy, ranked seventh globally and fourth in Europe among all cancers in women [1,2]. In Poland, a 2.5-fold increase in the incidence of EC has been observed in the last three decades, and in 2016, it accounted for almost 8% of cancer cases in women [3]. Obesity is a significant risk factor for EC, contributing to its high prevalence in developed countries [2,4]. An early diagnosis is associated with greater chances of survival (up to 90%) [5]. The most frequent and early symptom of EC is abnormal uterine bleeding after menopause [6]. Most cases of EC occur in the postmenopausal period, between 65 and 75 years of age [1,2]. However, 4% of women affected by EC are younger than 40 and require fertility-sparing treatment [7].

According to the classic model introduced by Bokhman in 1983, EC is classified into two types [8]. The more common type I—endometrioid adenocarcinoma—is composed of moderately or highly differentiated cells and is caused by hyperestrogenism. Type II EC is associated with an atrophic endometrium consisting of poorly differentiated cells. In contrast to type I EC, estrogen-independent type II tumors tend to metastasize and are diagnosed in the advanced stages of the disease, which worsens the prognosis [5]. However, this traditional classification system has some limitations, as highlighted by new molecular data [5]. Surgery is the first treatment for almost all women with EC [6]. The preoperative assessment of the histological type and stage of EC may help identify patients at risk of recurrence and define surgical management [2]. However, there is currently no reliable, noninvasive diagnostic method for the identification of early stages of the disease and the risk stratification of patients [1]. Moreover, there is no screening method for EC [1]. The use of prognostic biomarkers may fill this diagnostic gap [1,9].

The epithelial–mesenchymal transition (EMT), in which polarized epithelial cells undergo a change in signaling processes, gaining a phenotype typical of motile mesenchymal cells, has been considered a critical factor in EC progression and invasion [10,11]. EMT is characterized by so-called “cadherin switching”, that is, the downregulation of E-cadherin, the transmembrane glycoprotein responsible for cell–cell adhesion and maintenance of epithelial integrity, which is balanced by the increased expression of the mesenchymal marker, N-cadherin [11]. As a result, EC cells acquire the ability to migrate and proliferate [10]. The colonization of distant sites is possible through the reverse process, the mesenchymal–epithelial transition (MET) [12].

Low expression of E-cadherin and high expression of N-cadherin may act as an EMT marker and, consequently, as a marker of the occurrence of metastases. The level of E-cadherin was reduced in EC compared with the normal endometrium and showed a tendency to be lower in advanced-stage and histologically high-grade tumors [13]. Singh et al. [14] reported that high E-cadherin expression was associated with longer median survival and a reduced risk of disease progression and death in patients with stage IV or recurrent EC treated with tamoxifen and medroxyprogesterone. In turn, the elevated expression of N-cadherin in cancer indicates tumor aggressiveness and a poor patient prognosis. The serum level of soluble N-cadherin was shown to be higher in cancer patients than in healthy patients [15]. According to the study of Xie et al. [16], the expression of E-cadherin and N-cadherin in EC was related to tumor staging, cell differentiation degree, and depth of muscular layer infiltration, suggesting the usefulness of these markers for prognostic purposes. However, so far, cadherin levels have been assessed only in primary lesions.

This study aimed to evaluate the expression of E-cadherin and N-cadherin in primary endometrial lesions and the endocervix in patients with EC in search of noninvasive predictive factors. The impact of the tumor grade and the degree of cell differentiation on the process of cadherin switching and the correlation between the expression of E-cadherin and N-cadherin in the cervix and the primary tumor were also assessed.

## 2. Results

### 2.1. Patients’ Characteristics

The analysis included 101 patients with EC, of whom 8 (7.9%) were premenopausal and 93 (92.1%) were postmenopausal. The mean age of the patients was 66 ± 9.5 years (range: 45–86 years). The mean body mass index was 30.3 ± 5.3 (range: 19.2–43.9). The mean number of years since menopause was 14 ± 9.7. More than half of the patients gave birth two or three times (n = 60; 59.4%), while one in ten (n = 11; 10.9%) had never given birth. The detailed characteristics of the patients are shown in Table 1.

Most tumors were moderately (grade 2 [G2]; 43.6%) or poorly differentiated (G3; 43.6%) (Table 1). The FIGO clinical stages of EC were as follows: 16 patients had stage I EC; 11 patients had stage II; 7 patients had stage III; and 4 patients had stage IV. Due to the retrospective design of this study, it was not possible to determine the clinical stage in the remaining cases.

### 2.2. Immunohistochemical Expression of E-Cadherin and N-Cadherin

The expression levels of E-cadherin and N-cadherin in the tumor tissues of patients with EC are shown in Table 2. High expression of N-cadherin was determined more frequently than that of E-cadherin (86% vs. 57%). In over 43% of cases, low expression of E-cadherin was observed, which reflects changes during the EMT process. The differences in staining intensity are presented in Figure 1, Figure 2 and Figure 3.

### 2.3. E-Cadherin and N-Cadherin Expression Depending on the Tumor Grade

The degree of histological tumor differentiation was significantly associated with E-cadherin expression (*p* = 0.04). Low protein expression was more common in G2 tumors (57%), while high expression predominated in poorly differentiated G3 cancer cells (73%). There was no association between tumor grade and N-cadherin expression (*p* > 0.05). The data are presented in Table 3.

### 2.4. E-Cadherin and N-Cadherin Expression Depending on the Tissue Location

The expression of cadherins in type I EC was not associated with the tissue location (Table 4). Intense staining in EC cells was observed more frequently for N-cadherin than for E-cadherin (84% vs. 39%). The endocervix was characterized mainly by the low expression of both cadherins. In 17 patients (33%), high expression of N-cadherin was observed both in the endometrium and the endocervix, which may suggest the ongoing EMT process in two locations. Intense immunoreactions in EC cells were confirmed by a similarly high marker expression in the endocervix in only 4 of the 20 cases for E-cadherin (20%) and in 17 of the 43 cases (40%) for N-cadherin.

In type II EC, there were significant differences in cadherin expression between the tumor and the endocervix (Table 4). Intense immunoreactions with the tested markers were observed in most EC lesions (E-cadherin, 70%; N-cadherin, 76%). Over 92% of strongly stained endocervical preparations (E-cadherin, 22/24; N-cadherin, 26/27) showed high cadherin expression both in the endometrium and the endocervix. In the absence of strong immunoreactions in the endocervical tissue, high cadherin expression was observed in almost half of the cases in the EC cells (E-cadherin, 13/26; N-cadherin, 12/23), which may indicate the intensification of the EMT process only in the tumor tissue.

### 2.5. E-Cadherin and N-Cadherin Expression Depending on the Type of Endometrial Cancer

Significant differences were found between the expression of E-cadherin in the endocervix (*p* < 0.001) and the tumor (*p* = 0.001), depending on the type of EC (Table 4). In type I EC, both the endocervix and the tumor were characterized by low E-cadherin expression (86% and 61%, respectively). On the other hand, in type II EC, high E-cadherin expression was noted in almost half of the cases in the endocervix and in 70% of cases in the tumor. The expression of N-cadherin in the endocervix and the tumor did not differ significantly depending on the type of EC. In both types of cancer, high N-cadherin expression was observed in the endometrium (84% vs. 76%).

### 2.6. The Relationship between E-Cadherin and N-Cadherin Expression

In type I EC, no significant differences were found between E-cadherin and N-cadherin expression in the endocervix (Table 5). In most cases, low cadherin expression was noted (E-cadherin, 86%; N-cadherin, 61%). More than half of the cases (4/7) showing intense immunoreactions with E-cadherin antibodies in the endocervix were associated with high N-cadherin expression.

In type II EC, the expressions of both cadherins in the endocervix were corresponding (*p* = 0.02). In 17 of the 24 cases (71%), high E-cadherin expression was also associated with increased N-cadherin expression. On the other hand, 16 of the 26 cases (62%) with low E-cadherin expression also had low N-cadherin expression.

In both types of EC, there was no correlation between the expression of E-cadherin and N-cadherin in the tumor. In type I cancer, 19 of the 20 cases (95%) with high E-cadherin expression in the endometrium also showed high N-cadherin expression. However, for 24 of the 31 preparations (77%) that showed weak immunoreactions with the E-cadherin antibody, high N-cadherin expression in the tumor was determined. Similar findings were observed for type II EC.

## 3. Discussion

Endometrial cancer is one of the most common gynecologic malignancies; however, despite its prevalence, reliable noninvasive markers for early detection and prognostic evaluation remain elusive, posing challenges to timely diagnosis and effective management strategies. The biological material collected from the cervix of patients with EC would enable an easier, noninvasive assessment of the expression of potential markers. However, the conflicting results of the studies underscore the pressing need for further research and development of innovative biomarkers to improve outcomes in endometrial cancer patients. Several markers have been explored, including E-cadherin, N-cadherin, and cancer antigen 125 (CA-125), among others. According to Calis et al. [17], the level of cancer antigen 125 (used as a surface marker for ovarian cancer) in cervicovaginal secretion may have potential in the noninvasive screening of EC. Endometrial precancer, or cancer, was detected with a sensitivity of 78% and a specificity of 57% [17]. Because CA-125 levels may be elevated in numerous benign conditions, such as adenomyosis, uterine fibroids, and endometriosis, the determination of CA-125 alone seems to be insufficient. In turn, Costas et al. [18] developed an algorithm to identify patients with EC using genetic molecular tests. Targeting the 50 most representative point mutations resulted in a sensitivity of over 80%, although at a high cost and requiring a technology able to sequence large DNA regions [18]. 

The role of both cadherins in EC as specific regulators of EMT has been widely described [16,19,20,21,22,23]. As EMT is more pronounced in advanced disease [24], cadherin levels may serve as a predictive factor. However, data on their expression in various types of EC are lacking, especially for N-cadherin, as well as correlations of E-cadherin and N-cadherin levels between the primary tumor and the endocervix. In this study, the expression of E-cadherin and N-cadherin in the endocervix of patients with EC was investigated to assess the potential use of these cadherins as prognostic biomarkers. We observed high expression of N-cadherin more frequently than that of E-cadherin. Moreover, low expression of E-cadherin was found in more than 40% of samples. The composition of cadherins varies throughout the progression of tumors and their spread to other parts of the body in reaction to varying microenvironments. Decreased levels of E-cadherin result in the weakening of adherens junctions crucial for cell–cell adhesion, thereby promoting the detachment of cells from primary tumors. Conversely, N-cadherin has been noted to encourage cell aggregation and coordinated migration [25,26]. Moreover, it was demonstrated to enhance motility in human cancer cell lines. Tumor cells expressing N-cadherin may also potentially stimulate angiogenesis by interacting with vascular endothelial cells expressing N-cadherin throughout the progression of tumors [27]. High N-cadherin expression was suggested to increase the ability of tumor cells to metastasize to distant sites, overcoming the tumor-inhibitory effect of E-cadherin [15]. Therefore, high expression of N-cadherin promotes aggressive behavior of the cancer cells, and low expression of protective E-cadherin (44% of cases) confirms the ongoing EMT process in the analyzed samples. Yadav et al. [28] also found that the downregulation of E-cadherin (69% of cases) was associated with the upregulation of N-cadherin (82% of cases) in patients with EC. This is also in line with other studies showing a significant reduction in E-cadherin levels in EC compared with the proliferative endometrium [29]. This decline in E-cadherin expression is often accompanied by the upregulation of non-epithelial cadherins like N-cadherin and cadherin-11 during epithelial-mesenchymal transition (EMT), a phenomenon termed ‘cadherin switching’. Additionally, cells undergoing EMT acquire mesenchymal markers such as vimentin, fibronectin, and SPARC, among others, facilitating their interaction with the stroma, thereby promoting invasion and metastasis. Throughout tumor progression, E-cadherin can be functionally impaired via various mechanisms, including somatic mutation, downregulation of gene expression through promoter methylation, and/or transcriptional repression [28].

We also found that the degree of histological tumor differentiation affects E-cadherin expression in EC. The increased E-cadherin level in low-differentiated G3 cancer, along with the accompanying high N-cadherin expression, may suggest the ongoing intensive EMT process and the simultaneous reverse MET process. However, this observation is in contrast with the findings of Tanaka et al. [24], who reported that a lower degree of tumor differentiation was associated with reduced E-cadherin levels. Also, Lizawati et al. [30] found that tumor grade was the main predictor of downregulated E-cadherin expression, while Mell et al. [31] described a negative correlation of E-cadherin staining with the tumor grade, which was an independent predictor for EC progression and mortality. In addition, according to Youssef et al. [32], high E-cadherin expression occurred mainly in G1 tumors (77.8%). It was suggested that the expression of E-cadherin in high-grade EC is strongly associated with histological subtypes (serous carcinomas vs. poorly differentiated G3 endometrioid adenocarcinomas) [33]. 

The involvement of the endocervix in EC presents an intriguing aspect of disease progression. Studies have shown that abnormal cervical cytology is prevalent in EC patients, particularly in those with serous EC. These abnormalities are often associated with extrauterine disease or cervical involvement, which significantly worsens the prognosis. In a retrospective study by Roelofsen et al. [34], abnormal cervical cytology was found in 87.5% of women with serous EC and 37.8% of women with endometrioid EC. However, the sensitivity of the Papanicolaou test for EC is only about 40%, which limits the use of cervical cytology as a screening test [35]. In our study, the presence of cadherins was confirmed in both the primary tumor and the endocervix. We found that the expression of cadherins in type I EC was not significantly associated with tissue location. In type I EC, both the endocervix and the tumor were characterized by low E-cadherin expression. In this group of patients, the cancer process may occur evenly at the primary site and in the cervix. This is in contrast to the study by Rubeša et al. [11], who indicated rather stronger E-cadherin staining in type I tumors. In type II EC, the expression of both cadherins in the tumor differed significantly from their expression in the endocervix. Interestingly, in type II EC, in the absence of positive immunoreactions in the material obtained from the cervix, high cadherin expression in the tumor was observed in almost half of the cases, which suggests greater dynamics of the EMT process in the tumor than in the cervix.

In cervical cancer, the downregulation of E-cadherin was suggested as a diagnostic biomarker, indicating worsened cervical lesions [36]. The results of another study showed that only E-cadherin and P-cadherin immunochemistry may help predict prognosis in patients with early-stage cervical squamous cell carcinoma because of the very low expression of N-cadherin in cervical lesions, which is not associated with patient survival [37]. Similarly, although E-cadherin expression decreased and N-cadherin expression increased during the progression of cervical squamous cell lesions, N-cadherin was not an independent prognostic biomarker of early-stage squamous cervical cancer, in contrast to E-cadherin [37]. The determination of E-cadherin expression in breast cancer was useful in differentiating tumor subtypes, but it was not correlated with prognostic variables [38,39]. However, Yang et al. [40] suggested that E-cadherin could be a diagnostic biomarker in patients with lymph node metastasis and triple-negative breast cancer. 

In our study, we used the classic Bokhman’s dualistic model, based on the histological morphology of EC, because the molecular classification of EC was not performed as a standard at that time. Recently, the molecular classification of EC has been shown to improve and individualize risk-stratified patient management [41]. In 2013, the Cancer Genome Atlas proposed four categories of EC: POLE ultramutated, microsatellite instability hypermutated (loss of nuclear expression of one or more MMR proteins), copy-number low (nonspecific molecular profile), and copy-number high (p53 abnormal), which determine clinical outcomes [42,43,44]. A link between EMT and the molecular profile of EC is still poorly understood, except for p53 mutations. Recently, Ruan et al. [45] constructed a prognostic EMT-related gene signature for predicting the prognosis of EC patients. Higher copy number variations of *BRAF*, *KRAS*, *PIK3CA*, *PTEN*, and *TP53* and high mutation rates of *p53*, *PTEN*, and *KRAS* were identified in the high-risk group of EC patients. Mutations of the tumor suppressor p53, involved in cell cycle, apoptosis, autophagy, DNA repair, and interactions with immune cells, are commonly found in EC and implicated in the regulation of EMT [46]. Mutant p53 inhibits the transcription of miR-130b, promoting EMT in EC [47]. Moreover, higher p53 and Snail expression values were shown in high-grade EC [48].

Multiple studies have been conducted to ascertain serum-based diagnostic markers for endometriosis. Unfortunately, none of the suggested biomarkers, in isolation, have demonstrated clinically significant diagnostic specificity so far. Consequently, none of these biomarkers are currently integrated into routine clinical practices. An optimal diagnostic assay for endometriosis would exhibit elevated sensitivity and specificity, minimizing false negatives and false positives, ensuring accurate identification of patients with endometriosis, and avoiding unnecessary interventions for unaffected individuals. Overall, while noninvasive markers hold promise for enhancing EC detection and management, their effectiveness and reliability depend on various factors, including sensitivity, specificity, clinical utility, validation, integration with other modalities, and cost-effectiveness. Continued research and validation efforts are essential to optimize the use of these markers in clinical practice and improve patient care.

The limitation of this study is its retrospective design, which may introduce bias. Moreover, due to this study’s retrospective nature, it was impossible to determine the FIGO stage of EC in all patients. We collected the material when the molecular classification of EC was not a standard. Therefore, analysis in this respect was not reported to the Bioethics Committee and could not be performed. Perhaps the use of a new classification of EC will allow us to better correlate cancer development with cadherin metabolism.

## 4. Materials and Methods

### 4.1. Study Design and Tissue Samples

In this single-center retrospective study, we collected data on 101 patients with type I and II EC (International Federation of Gynecology and Obstetrics [FIGO] stages I–IV [49]) who had received curative surgery at the 1st Chair and Department of Oncological Gynaecology and Gynaecology at the Medical University in Lublin, Poland, from 2013 to 2018. Clinicopathological data were based on medical records. Routine formalin-fixed and paraffin-embedded material blocks were retrieved from the files of the Histopathology Laboratory at the Independent Public Clinical Hospital No. 1 in Lublin. For each patient, the sample of the primary tumor in the endometrium and the fragment of the endocervix were assessed. This study was approved by the Bioethics Committee at the Medical University in Lublin (approval no. KE-0254/212/2021).

### 4.2. Immunohistochemistry of E-Cadherin and N-Cadherin

Immunohistochemical analyses were performed following the reagent manufacturer’s protocol (Bios, Cambridge, MA, USA). Paraffin-embedded, 4 μm thick tissue sections placed on silanized glass slides (SuperFrost Plus, Thermo Scientific, Schwerte, Germany) were incubated at 60 °C for 45 min. The sections were dewaxed in xylene and rehydrated in a series of decreasing ethanol concentrations (100%, 95%, 70%, 50%) at room temperature for 3 min. For antigen retrieval, the sections were heated at 100 °C for 15 min in citrate buffer (pH = 6.0), set aside at room temperature for 20 min, and washed with Tris-buffered saline. The nonspecific binding was blocked by incubating the sections with diluted rabbit serum at room temperature for 120 min. To detect specific immunoreactivity, the sections were incubated with primary antibodies (1 µg/µL), E-cadherin Polyclonal Antibody (cat. no. bs-10009R, Bios USA), and N-cadherin Polyclonal Antibody (cat. no. bs-1172R, Bios USA) overnight at a temperature of −4 °C. Tissue specimens were then rinsed twice and incubated with secondary antibodies with gentle agitation for 120 min at room temperature, followed by applying a 3,3′-diaminobenzidine tetrahydrochloride solution. Then, the sections were washed, dehydrated with increasing concentrations of ethyl alcohol (80%, 95%, 100%), and cleared with xylene. The immunostained slides were covered with a coverslip for immediate microscopic evaluation. All laboratory procedures were performed strictly according to the manufacturer’s instructions.

### 4.3. Evaluation of Immunoreactivity

Two pathologists observed immunohistochemical E- and N-cadherin expression using five fields of view and a magnification of 200×. The intensity of staining was classified into four grades: (1) no immunoreaction—positivity in <10% of cancer cells; (2) weak immunoreaction—positivity in 11% to 25% of cancer cells; (3) moderate immunoreaction—positivity in 26% to 50% of cancer cells; (4) strong immunoreaction—positivity in more than 50% of cancer cells. For statistical analysis, low expression was defined as no immunoreaction or immunoreaction present in less than 26% of cancer cells, and high expression as immunoreaction in 26% of cancer cells or more.

### 4.4. Statistical Analysis

All data were subjected to statistical analysis, which was performed using STATISTICA v. 13.0 (Data Analysis Software System, StatSoft Inc., Tulsa, OK, USA). The values of the analyzed parameters measured on a nominal or ordinal scale were characterized by the frequency and percentage, and those measured on an interval scale by the arithmetic mean, standard deviation, and range of variation, depending on the distribution. Contingency tables and the χ^2^ test for independence and homogeneity were used to determine relationships between categorical variables. For the small sample sizes, the Yates correction was applied. A *p*-value of less than 0.05 was considered significant.

## 5. Conclusions

Based on the differences in E-cadherin and N-cadherin expression, the results of our study confirm “cadherin switching” related to the EMT, depending on the tumor grade, location, and type of EC. Both cadherins were detected not only in the primary tumor but also in the endocervix of patients with EC. The expression of both E-cadherin and N-cadherin, which can be determined noninvasively in cervical material, can be a valuable biomarker for EC and has the potential to be used in screening and predicting the clinical course of the disease.

## Figures and Tables

**Figure 1 ijms-25-03547-f001:**
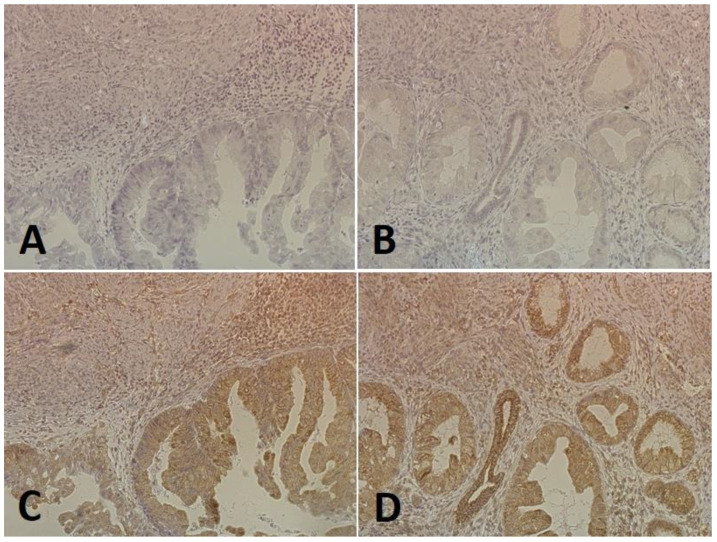
An example of a low expression of E-cadherin (**A**,**B**) and a high expression of N-cadherin (**C**,**D**) in the same tumor-containing fragment in a patient with endometrial cancer (magnification 200×).

**Figure 2 ijms-25-03547-f002:**
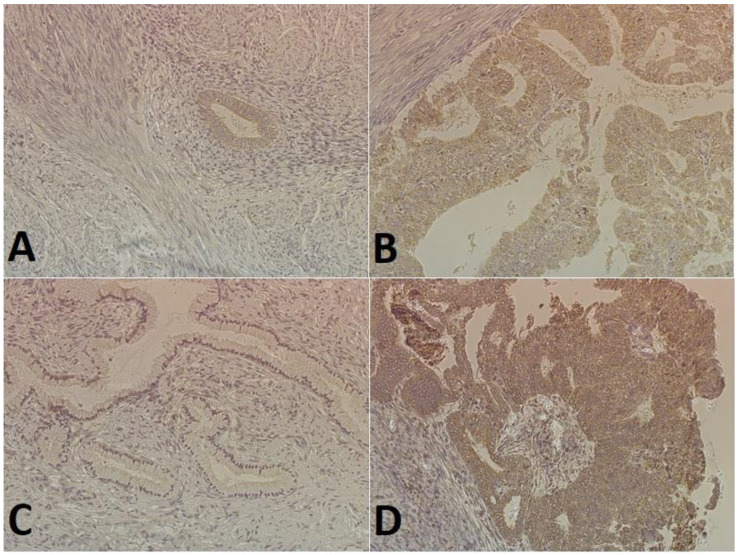
The staining of cadherins in endometrial cancer cells (magnification 200×): (**A**) low E-cadherin expression (weak staining in the cytoplasm of glandular epithelial cells); (**B**) high E-cadherin expression (intense staining in the cytoplasm of most cells); (**C**) no N-cadherin expression (glandular epithelial cells and the tumor stroma); (**D**) high expression N-cadherin (intense dark brown staining in the cytoplasm of some cells).

**Figure 3 ijms-25-03547-f003:**
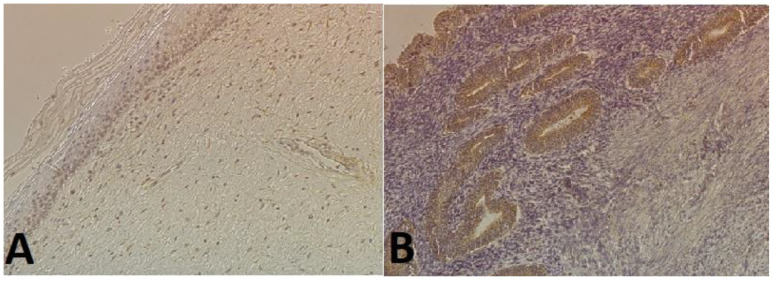
Examples of immunohistochemical staining with anti-E-cadherin antibody (magnification 200×): (**A**) in the endocervix (weak staining in the nuclei of epithelial cells); (**B**) in endometrial cancer cells (intense staining in glandular epithelial cells).

**Table 1 ijms-25-03547-t001:** Basic characteristics of the patients (n = 101). Data are presented as means ± SD (min–max) or n (%). BMI—body mass index.

Parameter	Value
Age (years)	66 ± 9.47 (45–86)
BMI (kg/m^2^)	30.3 ± 5.25 (19.2–43.9)
Menopausal status	
Years since menopause	14 ± 9.69 (0–38)
Premenopausal	8 (7.9)
Postmenopausal	93 (92.1)
Number of deliveries	
0	11 (10.9)
1	19 (18.8)
2	40 (39.6)
3	20 (19.8)
4	6 (5.9)
5	3 (2.9)
6	1 (1.0)
8	1 (1.0)
Tumor grade	
G1	4 (3.9)
G2	53 (52.5)
G3	44 (43.6)
Clinical stage	
IA	8 (7.9)
IB	8 (7.9)
II	11 (10.8)
IIIA	4 (3.9)
IIIB	3 (2.9)
IVB	4 (3.9)

**Table 2 ijms-25-03547-t002:** Expression levels of E-cadherin and N-cadherin in the tumor tissues of patients with endometrial cancer (n = 101).

Expression	Intensity of Immunoreaction	E-Cadherin n (%)	N-Cadherin n (%)
Low	No	4 (3.9)	2 (1.9)
	Weak	40 (39.6)	13 (12.8)
High	Moderate	30 (29.7)	20 (19.8)
	Strong	27 (26.7)	66 (65.3)

**Table 3 ijms-25-03547-t003:** Impact of histological differentiation (tumor grade) on the expression of N- and E-cadherin in endometrial cancer cells (n = 101). Data are presented as n (%). * Chi-Square test, *p*-values < 0.05 were considered statistically significant.

Tumor Grade	Expression					
	E-Cadherin			N-Cadherin		
	Low	High	*p*-Value *	Low	High	*p*-Value *
G1 (n = 4)	2 (50)	2 (50)	0.04	0 (0)	4 (100)	0.5
G2 (n = 53)	30 (57)	23 (43)		7 (13)	46 (87)	
G3 (n = 44)	12 (27)	32 (73)		8 (18)	36 (82)	

**Table 4 ijms-25-03547-t004:** Comparison of E-cadherin and N-cadherin expression depending on the tissue location (tumor vs. the endocervix) and histological type of endometrial cancer (type I vs. II). Data are presented as n (%). * χ^2^ test, *p*-values < 0.05 were considered statistically significant; p1—*p*-value for cadherin expression depending on the location; p2—*p*-value for cadherin expression in the endocervix depending on the type of endometrial cancer; p3—*p*-value for cadherin expression in the tumor depending on the type of endometrial cancer.

Endometrial Cancer	Expression in the Tumor	Expression in the Endocervix									
		E-Cadherin					N-Cadherin				
		Low	High	p1 *	p2 *	p3 *	Low	High	p1 *	p2 *	p3 *
Type I	Low	28 (55)	3 (6)	0.29	<0.001	0.001	4 (8)	4 (8)	0.59	0.19	0.29
	High	16 (31)	4 (8)				26 (51)	17 (33)			
Type II	Low	13 (26)	2 (4)	0.01			11 (22)	1 (2)	<0.001		
	High	13 (26)	22 (44)				12 (24)	26 (52)			

**Table 5 ijms-25-03547-t005:** Comparison of E-cadherin and N-cadherin expression in the tumor tissue and the endocervix in type I and II endometrial cancer. Data are presented as n (%). * χ^2^ test, *p*-values < 0.05 were considered statistically significant.

Endometrial Cancer	Expression of E-Cadherin	Expression of N-Cadherin					
		Endocervix			Tumor		
		Low	High	*p*-Value *	Low	High	*p*-Value *
Type I	Low	27 (53)	17 (33)	0.35	7 (14)	24 (47)	0.09
	High	3 (6)	4 (8)		1 (2)	19 (37)	
Type II	Low	16 (32)	10 (20)	0.02	4 (8)	11 (22)	0.77
	High	7 (14)	17 (34)		8 (16)	27 (54)	

## Data Availability

The data presented in this study are available on request from the corresponding author.

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
