# Peer review of "Expression of E-Cadherin and N-Cadherin in the Endocervix as a Predictive Factor in Patients with Endometrial Cancer"

_ijms, 2024, doi:10.3390/ijms25063547_

Round 1
Reviewer 1 Report
Comments and Suggestions for Authors
The authors conducted a fairly large amount of research and honestly described the results obtained. Unfortunately, the value of the work is reduced due to the peculiarities of the sample - the clinical stage of the disease, one of the most important parameters in such studies, was known only for 38 out of 101 patients and was not analyzed at all in this work. The second disadvantage of the work can be considered the lack of information about the expression of E and N-cadherins in endocervix in the control group, in patients without EC.
The discussion of the results is poorly structured and needs to be reworked. In addition, EMT/MET is not the only cadherin–dependent mechanism. Vascularization and activation of a number of signaling pathways may also play a role.
The slightly unexpected results associated with increased expression of E-cadherin in low-grade tumors are in no way a disadvantage of the work. Have the authors tried to compare the expression of this marker in groups divided by two criteria: EC I or II and Grade 1-2 or 3?
Finally, if we are talking about non-invasive markers, it is worth evaluating their effectiveness, sensitivity and specificity, there is all the data for this.
Comments on the text.
1. The caption in Figure 1 is not quite correct – it should probably be “An example of a low expression of E-cadherin and high expression of N-cadherin...”
2. In some cases, there is no indication of the fabric for which the data is presented (line 119, table 5, column 2)
Author Response
We would like to thank for your valuable comments which helped to improve this manuscript. Your suggestion was taken into consideration and appropriate information was provided. New/corrected parts are made in the track changes mode to facilitate the assessment of changes. We did our best to fulfil your expectations and we hope that you will be satisfied with our corrections.
Comments and Suggestions for Authors
The authors conducted a fairly large amount of research and honestly described the results obtained. Unfortunately, the value of the work is reduced due to the peculiarities of the sample - the clinical stage of the disease, one of the most important parameters in such studies, was known only for 38 out of 101 patients and was not analyzed at all in this work.
Response: We are aware of this drawback, however, due to the retrospective design of the study, we could not determine the clinical stage in the remaining cases. This information was included in the limitations paragraph.
The second disadvantage of the work can be considered the lack of information about the expression of E and N-cadherins in endocervix in the control group, in patients without EC.
Response: The aim of this study was to compare the expression of cadherin E and N in tumor and endocervix sample. However, we plan to compare the expression of these two molecules in specimens from patients with and without EC.
The discussion of the results is poorly structured and needs to be reworked. In addition, EMT/MET is not the only cadherin–dependent mechanism. Vascularization and activation of a number of signaling pathways may also play a role.
Response: The discussion was reorganized and additional information on the role of cadherin in vascularization and activation of signaling pathways was provided
The slightly unexpected results associated with increased expression of E-cadherin in low-grade tumors are in no way a disadvantage of the work. Have the authors tried to compare the expression of this marker in groups divided by two criteria: EC I or II and Grade 1-2 or 3?
Response: Yes, we compared the expression of both cadherins in type I and type II endometrial cancer. The results are presented in Table 4
Finally, if we are talking about non-invasive markers, it is worth evaluating their effectiveness, sensitivity and specificity, there is all the data for this.
Response: We provided short information concerning the specificity, effectiveness and reliability of available markers
Comments on the text.
- The caption in Figure 1 is not quite correct – it should probably be “An example of a low expression of E-cadherin and high expression of N-cadherin...”
Response: the capture was corrected
- In some cases, there is no indication of the fabric for which the data is presented (line 119, table 5, column 2)
Response: Line 119- endometrial cancer cells, table 5- tumor tissue and endocervix, column 2- not in the article. Unfortunately, we are not sure whether we understood the comments clearly. If not, please specify what data you are referring to.
Reviewer 2 Report
Comments and Suggestions for Authors
The article titled "Expression of E-cadherin and N-cadherin in the endocervix as a predictive factor in patients with endometrial cancer" presents a significant contribution to the literature on endometrial cancer.
Endometrial cancer (EC) stands as the most prevalent gynecological malignancy. This study evaluates the expression of E-cadherin and N-cadherin in primary endometrial lesions and the endocervix in patients with EC, aiming to identify noninvasive predictive factors.
The role of both cadherins in EC as specific regulators of epithelial-mesenchymal transition (EMT) has been extensively discussed. However, data on their expression in various types of EC remain limited, particularly for N-cadherin. High N-cadherin expression augments tumor cell metastatic potential to distant sites, counteracting the tumor-inhibitory effects of E-cadherin.
Given that EMT is more pronounced in advanced disease stages, cadherin levels may serve as predictive factors. Utilizing biological material collected from the cervix of EC patients could facilitate a simpler, noninvasive assessment of both potential markers. Nonetheless, correlations of E-cadherin and N-cadherin levels between the primary tumor and the endocervix have yet to be described.
This study delves into the expression of E-cadherin and N-cadherin in the endocervix of EC patients to evaluate the potential use of these cadherins as prognostic indicators. The presence of cadherins was confirmed in both the primary tumor and the endocervix. High N-cadherin expression (86% of cases), associated with aggressive cancer cell behavior, and low E-cadherin expression (44% of cases) affirm the ongoing EMT process. Furthermore, the degree of histological tumor differentiation influences E-cadherin expression in EC. Elevated E-cadherin levels in low-differentiated G3 cancers, alongside concurrent high N-cadherin expression, may suggest an intense ongoing EMT process and simultaneous reverse mesenchymal-epithelial transition (MET) process.
Based on disparities in E-cadherin and N-cadherin expression, the results validate "cadherin switching" linked to EMT, contingent upon tumor grade, location, and EC type. Both cadherins were identified not only in the primary tumor but also in the endocervix of EC patients.
In summary, the study proposes that the expression of both E-cadherin and N-cadherin, determinable noninvasively in cervical material, can serve as valuable biomarkers for EC and potentially aid in screening and predicting the clinical trajectory of the disease. The manuscript is well-articulated and draws reasonable conclusions.
Author Response
We would like to thank you for the appreciation of our work.